# Novel Photocatalytic Nanocomposite Made of Polymeric Carbon Nitride and Metal Oxide Nanoparticles

**DOI:** 10.3390/molecules24050874

**Published:** 2019-03-01

**Authors:** Iwona Koltsov, Jacek Wojnarowicz, Piotr Nyga, Julita Smalc-Koziorowska, Svitlana Stelmakh, Aleksandra Babyszko, Antoni W. Morawski, Witold Lojkowski

**Affiliations:** 1Institute of High Pressure Physics, Polish Academy of Sciences, Sokolowska 29/37, 01-142 Warsaw, Poland; j.wojnarowicz@labnano.pl (J.W.); julita.smalc.koziorowska@unipress.waw.pl (J.S.-K.); svetlana.stelmakh@unipress.waw.pl (S.S.); w.lojkowski@labnano.pl (W.L.); 2Military University of Technology, Institute of Optoelectronics, Urbanowicza 2, 00-908 Warsaw, Poland; piotr.nyga@wat.edu.pl; 3West Pomeranian University of Technology Szczecin, Pułaskiego 10, 70-322 Szczecin, Poland; ba08645@zut.edu.pl (A.B.); amor@zut.edu.pl (A.W.M.)

**Keywords:** microwave hydrothermal synthesis, AlOOH-ZrO_2_, nanocomposites, polymeric carbon nitride (PCN), band gap, photocatalysis, γ-Al_2_O_3_-ZrO_2_ nanopowders

## Abstract

Semiconducting polymers are promising materials for photocatalysis, batteries, fuel applications, etc. One of the most useful photocatalysts is polymeric carbon nitride (PCN), which is usually produced during melamine condensation. In this work, a novel method of obtaining a PCN nanocomposite, in which PCN forms an amorphous layer coating on oxide nanoparticles, is presented. Microwave hydrothermal synthesis (MHS) was used to synthesize a homogeneous mixture of nanoparticles consisting of 80 wt.% AlOOH and 20 wt.% of ZrO_2_. The nanopowders were mechanically milled with melamine, and the mixture was annealed in the temperature range of 400–600 °C with rapid heating and cooling. The above procedure lowers PCN formation to 400 °C. The following nanocomposite properties were investigated: band gap, specific surface area, particle size, morphology, phase composition, chemical composition, and photocatalytic activity. The specific surface of the PCN nanocomposite was as high as 70 m^2^/g, and the optical band gap was 3 eV. High photocatalytic activity in phenol degradation was observed. The proposed simple method, as well as the low-cost preparation procedure, permits the exploitation of PCN as a polymer semiconductor photocatalytic material.

## 1. Introduction

Photocatalysis is based on the photoabsorption by a semiconducting material at its surface. Photon absorption leads to the generation of electron–hole pairs, and the electron should reach the surface without hole–electron recombination. Therefore, considerable efforts have been made to enhance photocatalytic performance by effective charge separation. The major problems associated with a single organic component photocatalyst are as follows: (I) low specific surface area (SSA_BET_) of the photocatalyst materials, which limits the interactions of the organic molecules with the photocatalyst surface, and (II) poor stability and reusability [1]. In recent years, binary or ternary photocatalysts have attracted considerable attention owing to their favorable advanced oxidation processes under stimulated light [2,3]. Over the last few decades, numerous approaches have been developed to enhance the photocatalytic efficiency of a photocatalyst, including doping with various chemical elements (Cr, Fe, B, S) [4,5,6,7], incorporating co-catalysts [8], including oxides [9], or the coupling of two or more semiconductors. Some of these have shown promising photocatalytic activities for the degradation of toxic pollutants under visible light irradiation because of their structural features and suitable band gaps in the range of 1.7–3 eV. However, the challenge is to find a semiconductor photocatalyst that is synthesized in an energy-efficient way, does not use critical raw materials, and does not require complicated post-synthesis modifications to tune the band gap.

Recently, the attention of scientists has been focused on manufacturing composites and heterojunctions involving C/N/H-containing solids, especially on synthesizing a material called graphitic carbon nitride (g-C_3_N_4_) [10], polymeric carbon nitride (PCN), or melon [11]. This compound is a product of melamine thermal condensation, possesses an optical band gap of 2.7 eV, and has great chemical stability [10]. The polymeric organic semiconductor PCN was reported mainly as a favorable photocatalyst in water splitting for hydrogen production [10]. Due to its properties, polymeric carbon nitride has also been used in the fields of photocatalysis, electrochemistry, and photo-electrochemistry [10,11]. 

Despite various advantages, PCN is characterized by low photocatalytic activity, which is caused by the rapid recombination of photo-generated electron–hole pairs [12,13,14,15], and a low specific surface area (SSA_BET_). Thus, there is a need to improve its surface by various modifications and structural adjustments [16]. The introduction of a metal oxide (e.g., ZnO, TiO_2_, Al_2_O_3_, etc.) to PCN in order to form a heterostructured composite was found to be an effective method to enhance the long-term photostability and thermal stability [17]. However, the specific surface area of these composites is not high enough. PCN displays a SSA_BET_ of approximately, or even below, 10 m^2^/g [17]. To date, many complicated preparation methods have been developed to obtain PCN-based materials with larger SSA_BET_. In principle, these methods can be classified into two categories. One category involves changing the morphology (e.g., porous nanostructure, hollow spheres, etc.) of PCN by template-assisted methods [17,18,19,20,21,22]. Another category is top-down chemical or thermal exfoliation of bulk PCN [17]. However, the abovementioned methods need additional post-treatment processes. Obviously, a simple, direct, and template-free strategy to synthesize PCN-based materials with large SSA_BET_ is desired.

The aim of our work was to utilize AlOOH (i.e., boehmite, aluminum oxyhydroxide) and ZrO_2_ nanoparticles in microwave hydrothermal synthesis as components of a PCN nanocomposite. The role of microwave hydrothermal synthesis in the obtaining of AlOOH-ZrO_2_ nanopowders during co-synthesis is described in our previous works [23,24,25]. Briefly, the process ensures a narrow size distribution, a high degree of crystallinity, and uniform intermixing of the two nanopowders in nanoscale [23,24,26,27,28,29]. The justification for the use of homogeneous AlOOH-ZrO_2_ nanopowders as a substrate for the deposition of PCN is to provide increased specific surface area (SSA_BET_) and provide suitable thermal and mechanical stability. To the best of our knowledge, the role of homogeneous nanopowders of AlOOH-ZrO_2_ in PCN nanocomposite formation is investigated here for the first time. All the ingredients of the nanocomposite, AlOOH, ZrO_2_, and PCN are earth-abundant, which makes our PCN nanocomposite an ideal, inexpensive photocatalyst candidate. 

## 2. Results

### Nanocomposite Characterization

The starting homogeneous nanopowders of AlOOH-ZrO_2_ used in this work consisted of 80 wt.% of AlOOH and 20 wt.% of ZrO_2_ [23,24,25]. In order to mill it with melamine, 80 wt.% of the nanopowders and 20 wt.% of melamine were used. The mixture was annealed at 400, 450, 500, and 600 °C. The synthesized nanocomposites were cooled with reference to the annealing temperature for better clarity. 

Figure 1a shows the FTIR spectra for the nanocomposites manufactured in the temperature range of 400–600 °C. In this experiment, polymeric carbon nitride was identified only in the nanocomposite prepared at 400 °C. It has been reported that the absorption band in the 1420–1620 cm^−1^ region is assigned to CN stretching and −NH_2_ bending vibrations in melamine [16]. The broad peak at 3100–3400 cm^−1^ can be attributed to the stretching vibration of N–H and O–H. The other peaks at 1257 and 1326 cm^−1^ are present due to the stretching vibration of aromatic C–N heterocycles consisting of trigonal N–C_3_ and C–NH–C bridging units [16], representing the formation of C–N–C bonds. The main indication of melon in a sample is the presence of a band around 800 cm^−1^, which belongs to triazine ring vibration in PCN [16]. It can be seen that the higher the annealing temperature of the nanocomposites, the weaker the intensity of the C–N bands.

Figure 1b presents the X-ray powder diffraction (XRD) spectra of the nanocomposites annealed in the temperature range of 400–600 °C. Interestingly, XRD analysis does not show the presence of the AlOOH phase even in the nanocomposite annealed at 400 °C. The phase transition of AlOOH to γ-Al_2_O_3_ already took place [23]. Besides the γ-Al_2_O_3_ phase, all the samples contained ZrO_2_. In addition, the nanocomposite annealed at 450 °C contained traces of melamine condensation procucts. The lack of the PCN phase in the nanocomposite annealed at 400 °C may indicate an amorphous phase of this compound.

For the TEM examination, we chose only the composition that contains PCN (Figure 1a). Figure 2 shows the TEM images of the nanocomposite obtained at 400 °C. It is possible to distinguish three types of morphology: tetrahedral 30–50 nm flakes, which can be assigned to γ-Al_2_O_3_; spherical 3–5 nm ZrO_2_ particles; and 5–10 nm amorphous layers, which can be assigned to PCN (Figure 2b,d).

Table 1 presents the results of SSA_BET_, helium density, and the average particle size of the nanocomposites obtained at various temperatures. There is a visible difference between SSA_BET_ for the nanocomposite annealed at 400 °C and for the nanocomposites obtained at higher temperatures. 

The reference sample γ-Al_2_O_3_-ZrO_2_ nanopowder annealed at 400 °C is characterized by a SSA_BET_ of 108 m^2^/g. The presence of a thin, nanometric PCN layer on the nanoparticle surface slightly reduces SSA_BET_ (nanocomposite 400 °C, Table 1). On the other hand, the SSA_BET_ value of above 100 m^2^/g for the nanocomposites annealed at 450 and 500 °C could be the result of the crystallization of some of the melamine condensation products, which decompose at higher temperatures. It is worth noting that the pure γ-Al_2_O_3_-ZrO_2_ obtained after annealing at 400 °C (reference for PCN nanocomposite 400 °C) had an evident influence on the SSA_BET_ of the PCN nanocomposite (Table 1). The deposition of the PCN layer on Al_2_O_3_-ZrO_2_ reduced slightly the SSA_BET_ from 108 to ~80 m^2^/g. This value of SSA_BET_ for the PCN nanocomposite is relatively high, because pure PCN is characterized by a SSA_BET_ of ~10 m^2^/g [26].

Interestingly, the density values of the nanocomposites annealed at 400, 500, and 600 °C are nearly the same. Only one sample (the nanocomposite annealed at 450 °C) has a slightly lower density (Table 1). In addition, all the prepared nanocomposites showed smaller densities than the reference sample with a density of 3.75 g/cm^3^ (Al_2_O_3_-ZrO_2_ annealed at 400 °C). This fact could be explained by the reduction of Zr^4+^ and Al^3+^ ions in the presence of melamine ok its decomposition products, obtaining metallic precipitations not visible during XRD examination. Thus, these ions could lead to the increased density of the PCN nanocomposites in comparison with the reference sample.

The relationship between SSA_BET_ and the density values for the homogenous nanomaterials was observed in our prior research [27,28]: the higher the SSA_BET_, the smaller the density. Our PCN nanocomposites do not show the abovementioned trend (Table 1). In this case, the SSA_BET_ differs while the density remains approximately constant. We attribute that lack of significant change of the density to the applied annealing temperature, melamine condensation, and its interconversion processes. It means, for example, that for nanocomposites prepared at 450 °C, the nanoparticles and pores of Al_2_O_3_-ZrO_2_ are covered by a PCN layer (closed pores), which causes lower density (helium pycnometry). At higher temperatures, the PCN undergoes further interconversion and decomposition, which increases the SSA_BET_ while the density remains at the same value.

Table 1 also shows the average pore diameter and total pore volume for the investigated nanocomposites. There is no influence of porosity on the photocatalytic properties of the nanocomposites. Nanocomposite 400 °C has the smallest SSA_BET_ and porosity and shows the largest photocatalytic effect. He et al. [29], who investigated polyoxometalate (POMs)-functionalized g-C_3_N_4_, showed that, compared with C_3_N_4_, POMs-modified samples demonstrated slightly better efficiencies in photodegradation of phenol and Methylene blue (MB). It was suggested that the increased specific surface area, porous volume, and efficient charge transfer can influence photocatalysis [26]. 

In comparison to He et al. [26], our materials are characterized by higher specific surface areas and bigger pore volumes. Thus, we think that the active PCN layer on the surface of Al_2_O_3_-ZrO_2_ prepared at 400 °C has a greater effect on photocatalytic performance in our case than SSA_BET_ and the material’s pore volume.

Figure 3a shows the total reflectance of the nanocomposites prepared in the temperature range of 400–500 °C. As-synthesized (AS) AlOOH-ZrO_2_ homogenous nanopowders after annealing at 600 °C, where AlOOH transformed completely into γ-Al_2_O_3_, were added as a reference. 

The reflectance of all the samples in the longer wavelength area of the spectrum is relatively high. In order to calculate the material band gap, we used the procedure described in [30,31]. Briefly, the reflectance was converted into a Kubelka–Munk function using Equation (1) [32,33]:(1)$$F\left(R\right)=\frac{{\left(1-R\right)}^{2}}{2R}$$where F(R) is the Kubelka–Munk function and R is the sample reflectance.

Information about the band gap (Eg) can be obtained using the equation proposed by Tauc, Davis, and Mott, i.e., Equation (2) [34,35]:(2)$$\alpha h\nu =A{\;(h\nu -Eg)}^{1/n}$$where α is the absorption of the material, h is the Planck constant, ν the frequency of light, and Eg is the band gap. The exponent 1/n depends on the type of transition in the material, and for the cases of all the samples, we assumed a direct transition; thus, n = 2. Because F(R) is proportional to α [36], one can determine the band gap from the following formula:(3)$${\left(F\left(R\right)\;h\nu \right)}^{2}=A\;\left(h\nu -Eg\right)$$The band gap can be determined from a plot of (F(R) hν)^2^ against hν by extrapolating the F(R) value to zero (Figure 3b). Table 2 shows the calculated band gap values for the investigated materials.

Photocatalytic properties of the nanocomposites prepared at 400, 450, and 500 °C are presented in Figure 4 and Figure 5. In addition, the photocatalytic tests were also performed for a reference sample: AlOOH-ZrO_2_ nanopowders annealed at 400 °C for 5 h (after annealing, the nanopowders consisted of γ-Al_2_O_3_ and ZrO_2_). Figure 4 and Figure 5 show phenol and Orange II adsorption and decomposition in the presence of the mentioned nanocomposites, respectively.

Based on the analysis of the obtained results, it was found that for all the tested photocatalysts, the adsorption process took 1 h before the actual photocatalytic oxidation process took place. Figure 4a, where the adsorption of phenol is presented, shows that the most efficient nanocomposite was the one annealed at 400 °C. The aforementioned PCN nanocomposite adsorbed 60% of the phenol from the solution. The adsorption on the reference sample was negligible, and on the nanocomposites annealed at 450 and 500 °C, approximately 10% of the phenol was adsorbed.

Next, the solution was irradiated with UV light. The results of the phenol photocatalytic decomposition are shown in Figure 4b. In Figure 4b, the absolute change of the phenol concentration is shown. The first step of the process (1 h) occurred in the dark and corresponded to adsorption. After 1 h, the UV light was turned on and the photocatalytic process was conducted. As a result, the most efficient material was the nanocomposite annealed at 400 °C. There was no phenol decomposition on the reference material and very little phenol decomposition on the nanocomposites annealed at 450 and 500 °C, which is also shown in Figure 4c in the change of the relative concentration of phenol. The final degree of phenol decomposition for the PCN nanocomposite annealed at 400 °C was approximately 58%.

Another model pollutant used in the experiments was Orange II. The relative changes of the Orange II concentration during adsorption in the dark are presented in Figure 5a. The adsorption occurred the most efficiently on the nanocomposites annealed at 450 °C, but it was only on the level of 25%. For the other materials, the adsorption was less efficient.

In Figure 5b, the photocatalytic decomposition of Orange II is presented. After 1 h of adsorption in the dark, UV light was applied, and the decomposition of the pollutant started. At first, the decomposition was more efficient on the nanocomposites annealed at 450 °C, but after 3 h, the PCN nanocomposite annealed at 400 °C was the most effective, reaching (after 5 h) 43% Orange II decomposition.

## 3. Discussion

The results show that the most promising material for photocatalytic applications is γ-Al_2_O_3_-ZrO_2_ covered by a PCN layer (Figure 6) obtained at 400 °C. The temperature of 400 °C is unexpected for PCN formation, because melamine, according to the literature [37], undergoes a transition to PCN above 600 °C. In order to synthesize PCN, the annealing of melamine has to be precisely stopped at the right time and at a given temperature. According to [37], the decomposition of melamine takes place through dimer (melam) and then trimer (melem) formation at a temperature below 400 °C. Melem exists in the temperature range of 400–500 °C. Above 600 °C, melem starts to polymerize to PCN (melon) [37]. In our case, the formation of PCN took place at a ~200 °C lower temperature than reported in the literature. In our opinion, this is due to the presence of nano-sized metal oxide in the nanocomposite and a fast heating rate (50 °C/min). For comparison, the pure melamine decomposition temperature tested in our laboratory is the same as that reported in the literature [37], ~600 °C. The presence of uniform, co-synthesized AlOOH-ZrO_2_ nanopowder shifts the formation of PCN by 200 °C towards lower temperatures.

The very fast heating (50 °C/min) and long dwelling time at 400 °C (5h) applied in our synthesis process allows the covering of boehmite particles by an amorphous layer of PCN and allows AlOOH particles to transform into γ-Al_2_O_3_ without destroying the PCN layer (Figure 2d). We think that such phase transformation was possible without destroying the PCN layer, because it was a very slow process, which took place within the same crystal structure with cubic symmetry.

Covering γ-Al_2_O_3_ and ZrO_2_ nanoparticles with an amorphous layer of PCN in the nanocomposite annealed at 400 °C can be also confirmed by the SSA_BET_ results. Table 1 shows that the specific surface area of the PCN nanocomposite (400 °C) is slightly lower than that of the γ-Al_2_O_3_-ZrO_2_ nanopowder but much smaller than the SSA_BET_ of the nanocomposites prepared in the temperature range of 450–500 °C. This can be explained by the covering of the surface of the nanoparticles with a thin organic layer and the fact that some PCN was formed in the gaps between particles.

Another confirmation of the PCN layer formation is the band gap value, 3 eV, which was significantly lower than the reference material (pure γ-Al_2_O_3_-ZrO_2_ nanopowders annealed at 600 °C), characterized by 5.1 and 5.2 eV band gaps. The layer presumably screens the nanoparticles from interaction with light, and they do not contribute to the overall band gap. On the other hand, pure PCN has a moderate bandgap of 2.7 eV, corresponding to an optical wavelength of 460 nm [14]. Considering the thermodynamic losses and other potentials in the photocatalytic process, the band gap of 2.7 eV can be found in between 2 eV and 3.1 eV [14].

Polymeric carbon nitride materials have been investigated recently in various fields of photocatalysis [1,3,6,7]. Carbon nitrides have applications as sanitizers for the removal of contaminants, including pollutants and pathogenic microorganisms from drinking water and air [35]. There are also studies [1,26,38,39,40,41] in which PCN has been successfully used in the degradation of soluble dyes, such as methyl orange (MO), methylene blue (MB), phenol, rhodamine B, and crystal violet (CV). 

Vattikuti et al. [1] showed that MoS_2_/Al_2_O_3_/g-C_3_N_4_ nanocomposite has good photocatalytic performance for the degradation of CV dye under visible light irradiation. They reported [1] that the photodegradation rate of CV increased with the increasing Al_2_O_3_ and MoS_2_ content up to 20% in comparison with pure g-C_3_N_4_. The photocatalytic activity of this nanocomposite was approximately 10.28 higher than that of pure g-C_3_N_4_. On the other hand, He et al. [26] found that pure PCN was not as effective in phenol decomposition as a POMs-modified sample, where the concentration of phenol decreased up to ~0.5 over 3 h. This result was worse than that in our case where the PCN nanocomposite annealed at 400 °C showed a decrease of phenol concentration to 0.5 over 30 min (Figure 4c). However, our outcomes are in contrast to the results described by Ren et al. [40], who investigated the photocatalytic properties of Ag_2_O/g-C_3_N_4_ composites. They demonstrated high photocatalytic activity for phenol degradation [40]. They [40] showed that in Ag_2_O/g-C_3_N_4_ composites under UV- and visible-light irradiation, phenol completely degraded in 20 and 90 min, respectively. The improved photocatalytic activities were attributed to the formation of a heterostructure between Ag_2_O and g-C_3_N_4_, the strong visible-light absorption, and the high separation efficiency of photoinduced electron—hole pairs resulting from the highly dispersed Ag_2_O particles [40]. 

In general, our findings regarding the photocatalytic activity of PCN nanocomposite are in agreement with others [7,26,29,38,39,41]. We think that photocatalytic efficiency depends more on the type of composite with PCN, then its specific surface area or porous volume structure. The photocatalytic properties of the PCN nanocomposites presented in this manuscript were investigated for the first time. Thus, it is difficult to compare our results with others. In summary, the addition of PCN nanocomposites to the presence of organic dyes and pollutants leads to relatively fast degradation of the contaminants. 

## 4. Materials and Methods

The procedure for synthesizing nanopowders containing AlOOH with ZrO_2_ addition is described in detail elsewhere [23,24,25]. The reagents used in the process were zirconyl chloride octahydrate (ZrOCl_2_·8H_2_O, Sigma-Aldrich, St. Louis, MI, USA (99.5%)), sodium hydroxide (NaOH, analytically pure, Chempur, Piekary Sląskie, Poland), and aluminum nitrate nonahydrate (Al(NO_3_)_3_·9H_2_O, analytically pure, Chempur, Piekary Sląskie, Poland). The microwave reactions took place in a microwave reactor (2.45 GHz, 600 W, MAGNUM II ERTEC, Wrocław, Poland). As-prepared AlOOH-ZrO_2_ nanopowder was hand mixed in a zirconia mortar with 20 wt.% of melamine (Sigma-Aldrich, CAS Number 108-78-1 (99%)). In the next step, the prepared powder was annealed in a tube furnace (homemade, Model 1, IHPP PAS, Warsaw, Poland) in 4 different temperatures (in the range of 400–600 °C) for 5 h in air (Figure 7). In all cases, the constant heating rate of 50 °C/min was applied.

All the powders after synthesis were exanimated using a Fourier transform infrared (FTIR) spectrometer (Bruker Optics, Tensor 27, Bruker BioSpin GmbH, Rheinstetten, Germany) equipped with a diamond attenuated total reflectance (ATR) accessory. The ATR-FTIR spectra were recorded at room temperature in the 4000–400 cm^−1^ range. The spectral resolution and accuracy of the measurements were 4 cm^−1^ and 1 cm^−1^, respectively.

X-ray diffraction (XRD) patterns of the nanopowders were collected on a diffractometer (X’Pert PRO, PANalytical, Almelo, Netherlands) equipped with a copper anode (Cu K 1) and an ultra-fast PIXcel1D detector. The analysis was performed at room temperature in the 2θ range of 10–80° with a step size of 0.03°.

The microstructures of the nanopowders and nanocomposites were investigated using conventional high-resolution (HR) transmission electron microscopy (TEM) and scanning (STEM) techniques with a FEI TECNAI G2 F20 S-TWIN electron microscope (Thermo Fisher Scientific, Waltham, MA, USA).

Helium density measurements were carried out using a helium pycnometer (AccuPyc II 1340, FoamPyc V1.06, Micromeritics, Norcross, GA, USA). The measurements were carried out in accordance with the ISO 12154:2014 standard at 25 °C.

The specific surface area of the nanopowders was determined using a surface analyzer (Gemini 262 2360, V 2.01, Micromeritics, Norcross, GA, USA). The nitrogen adsorption method was applied based on the linear 263 form of the Brunauer–Emmett–Teller (BET) isotherm equation. The obtained data were analyzed using the MicroActive software V4.03 (Interactive Data Analysis Software, Micromeritics). The detailed experimental procedure and the determination of particle size using the SSA_BET_ method is described elsewhere [42]. Based on the results of the specific surface area and density, the average particle size was calculated with the assumption that all the particles were spherical and identical.

The band gap experiments were conducted for nanocomposites formed into pellets. The UV–Vis reflectance of samples was measured at room temperature using Lambda 650 (Perkin-Elmer, Waltham, MA, USA) equipped with an integrating sphere module. Spectralon SRS-99-01030 reflectance material was used as a reference sample. The total reflectance of the samples was collected.

The photocatalytic activity of the obtained nanomaterials was determined on the basis of water pollutant degradation tests under the influence of UV–Vis radiation with high UV intensity (6 lamps, 20 W each, 40 cm long, type: ISOLDE, Koninklijke Philips N.V., Amsterdam, Netherlands). The lamp used in the experiments was not monochromatic. The wavelength range was from 200–800 nm. Phenol (neutral) and Orange II dye (anionic dye) were used as the model pollutants. In a 150 cm^3^ glass beaker, 20 mg of the appropriate photocatalyst (nanocomposite) was suspended and then 100 cm^3^ of the dye or phenol solution with an initial concentration of 20 mg/dm^3^ was added. The final concentration of the catalyst in the mixture was 0,02 g/dm^3^. The suspension was stirred continuously using a magnetic stirrer (500 rpm). Further, suspensions were irradiated with ultraviolet light for 5 h. The pollutant degradation process consisted of 2 stages. The first stage was an adsorption in a darkroom conducted until the equilibrium between the liquid and solid was established. The optimal time for adsorption was determined individually for each compound. Samples were taken at identical time intervals (every hour) and tested on a UV–Vis spectrophotometer. The concentrations of the individual samples were determined on the basis of calibration curve equations drafted for the given pollutants.

## 5. Conclusions

In this work a PCN-γ-Al_2_O_3_-ZrO_2_ nanocomposite was obtained for the first time.

The synthesis of the PCN-based nanocomposite photocatalyst consisted of two steps: (a) microwave hydrothermal synthesis of AlOOH-ZrO_2_ nanopowders and (b) controlled addition of melamine to AlOOH-ZrO_2_ nanopowders and subsequent thermal treatment to obtain PCN-nanocomposites. This synthesis approach to PCN-nanocomposites is unique and produces PCN nanocomposite photocatalysts, which are highly in demand, in an easy, clean, energy-efficient, and low-cost way. 

It was shown that PCN was formed at the surface of γ-Al_2_O_3_ and ZrO_2_ at 400 °C by means of a melamine thermal condensation process, which took place at a ~200 °C lower temperature than the state of the art. We postulate that this was possibly due to the interaction of the melamine with the γ-Al_2_O_3_ and ZrO_2_ nanoparticles, which provide high specific surface areas and morphology favorable for the formation of a PCN layer. 

It was observed that AlOOH fully transformed into γ-Al_2_O_3_ without affecting the PCN layer structure.

The PCN nanocomposite has a 3 eV band gap and shows significant photocatalytic ability for common pollutant adsorption and degradation. The proposed simple and low-cost preparation method permits the exploitation of PCN as a polymer semiconductor photocatalytic material.

## Figures and Tables

**Figure 1 molecules-24-00874-f001:**
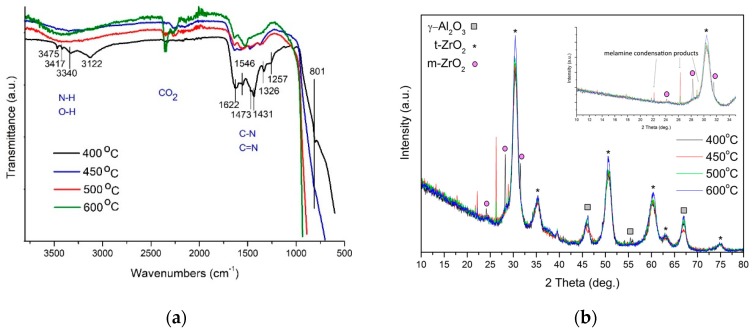
(**a**) FTIR spectra and (**b**) XRD spectra of the nanocomposites prepared in the temperature range 400–600 °C.

**Figure 2 molecules-24-00874-f002:**
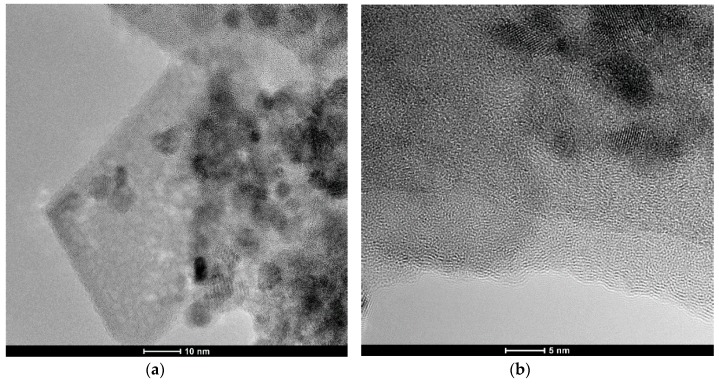
(**a**,**b**) TEM images of the nanocomposite obtained at 400 °C; (**c**,**d**) STEM images of the nanocomposite obtained at 400 °C.

**Figure 3 molecules-24-00874-f003:**
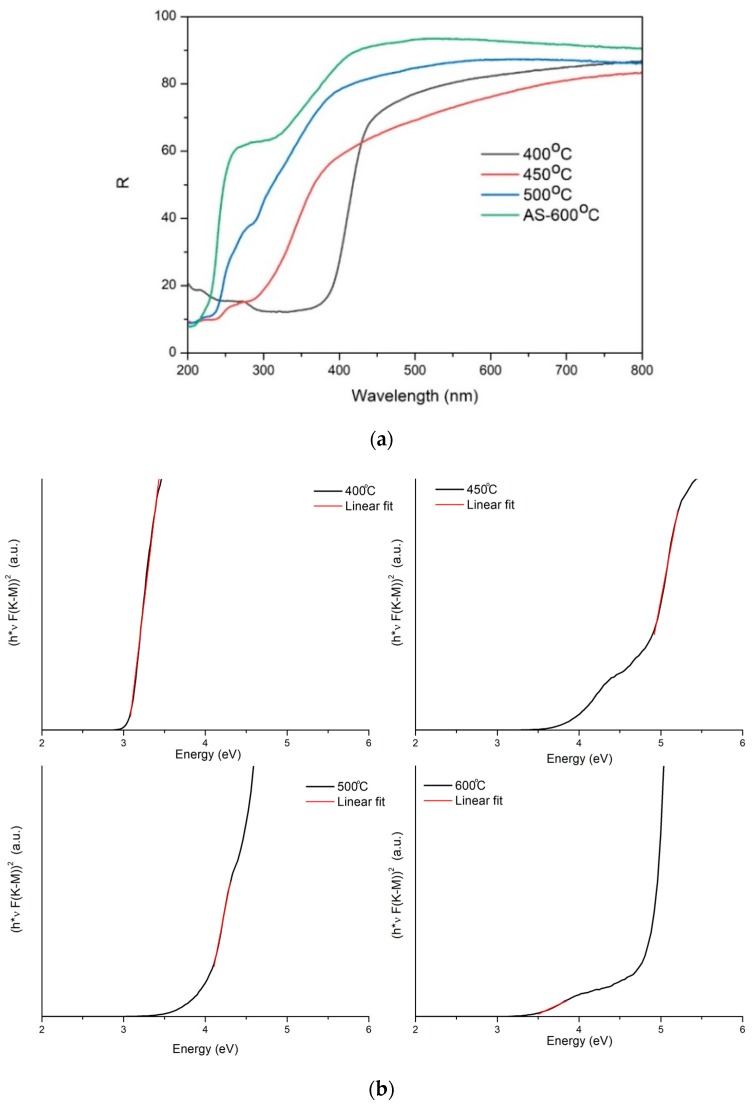
(**a**) Total reflectance of the nanocomposites prepared in the 400–500 °C temperature range. In addition, the AlOOH-ZrO_2_ nanopowders annealed at 600 °C were added as a reference (AS 600 °C, green line). (**b**) Reflectance UV–Vis spectra of Kubelka–Munk (K–M) functions vs energy (eV) for the investigated materials.

**Figure 4 molecules-24-00874-f004:**
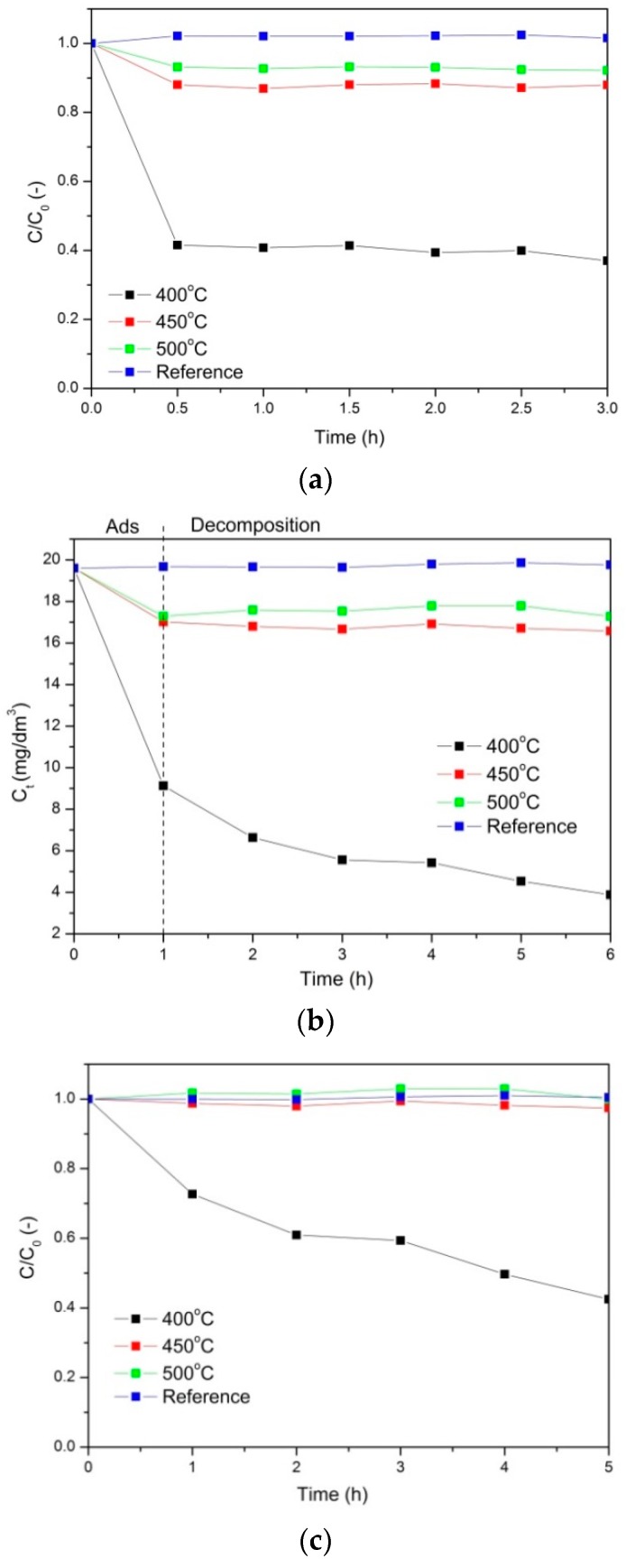
Adsorption of phenol (**a**) Photocatalytic decomposition of phenol, where (**b**) presents the absolute change of the phenol concentration and (**c**) presents the relative change of the phenol concentration.

**Figure 5 molecules-24-00874-f005:**
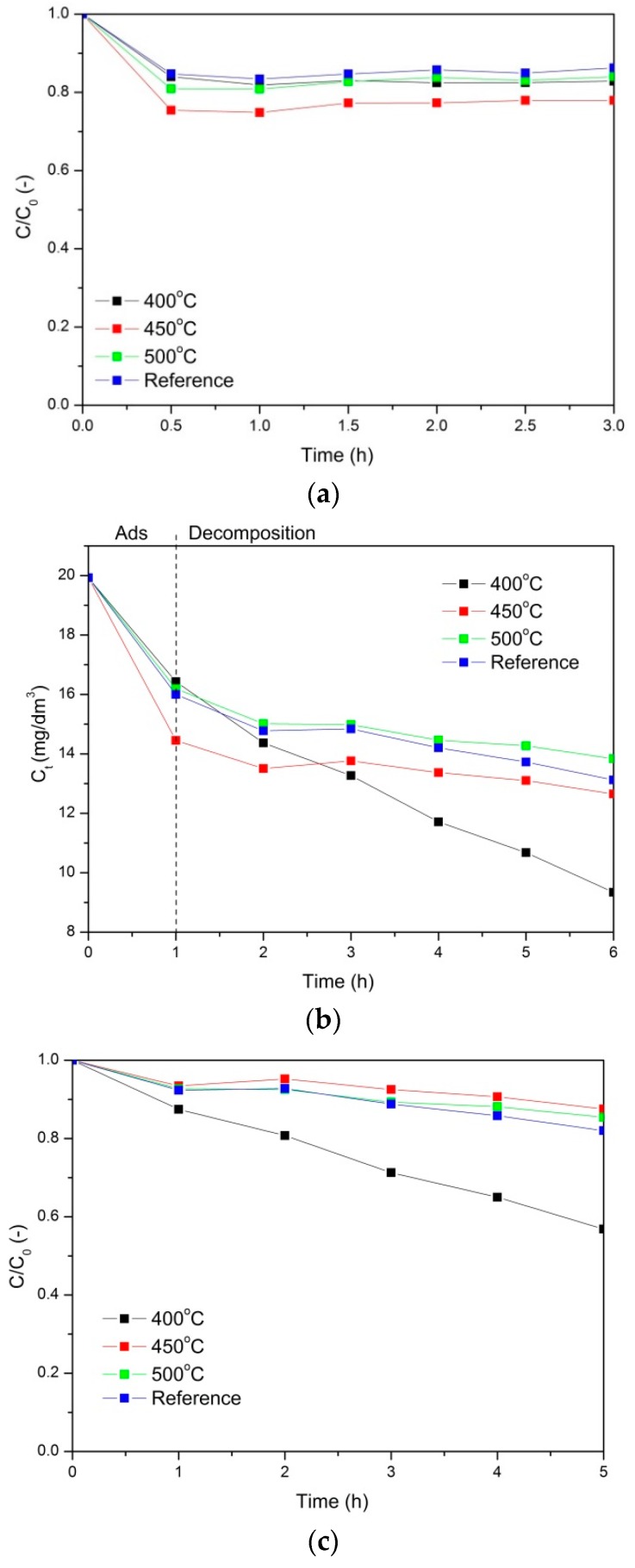
Adsorption of Orange II (**a**) Photocatalytic decomposition of Orange II, where (**b**) presents the absolute change of the phenol concentration and (**c**) presents the relative change of the phenol concentration.

**Figure 6 molecules-24-00874-f006:**
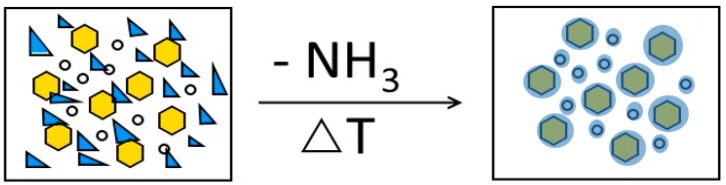
Scheme of AlOOH, ZrO_2_, and melamine leading to the formation of the polymeric carbon nitride (PCN) nanocomposite. The circles represent ZrO_2_, the pentagons represent AlOOH/Al_2_O_3_, and the blue triangles represent melamine powder.

**Figure 7 molecules-24-00874-f007:**
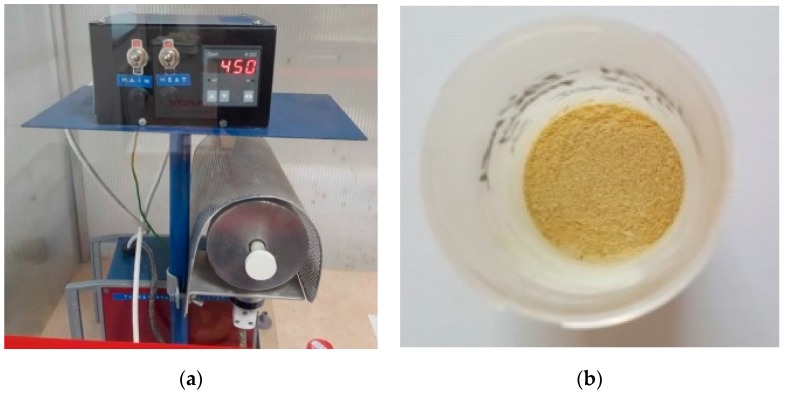
(**a**) Homemade tube furnace used for nanocomposites synthesis; (**b**) PCN nanocomposite obtained at 400 °C.

**Table 1 molecules-24-00874-t001:** Specific surface area, helium density, particle size, and pores dimensions of the investigated materials. The SSA_BET_ represents specific surface area, and BJH describes Barrett- Joyner-Halenda method.

Sample Name	SSA_BET_	Density (g/cm^3^)	Average Particle Size Calculated from SSA_BET_ (nm)	BJH Adsorption Average Pore Diameter (nm)	Total Pore Volume (cm^3^/g)
	Multipoint (m^2^/g)				
Nanocomposite 400 °C	79.5	3.75 ± 0.01	20	9.8	0.191
Reference (Al_2_O_3_-ZrO_2_ annealed at 400 °C)	108.6	3.51 ± 0.2	16	12.1	0.337
Nanocomposite 450 °C	132.9	3.56 ± 0.01	13	10.9	0.311
Nanocomposite 500 °C	120.0	3.71 ± 0.02	14	-	-
Nanocomposite 600 °C	92.4	3.76 ± 0.06	17	13.9	0.313

**Table 2 molecules-24-00874-t002:** Band gap values calculated for the investigated nanocomposites.

Sample Name	Band Gap (eV)
Nanocomposite 400 °C	3.0
Nanocomposite 450 °C	3.5/4.7
Nanocomposite 500 °C	4.0/4.8
AS 600 °C	5.1–5.2

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
