# Peer review of "Novel Photocatalytic Nanocomposite Made of Polymeric Carbon Nitride and Metal Oxide Nanoparticles"

_molecules, 2019, doi:10.3390/molecules24050874_

Reviewer 1 Report

The manuscript requires major revision prior to final publication decision.

See attached referee comments.

Author Response

Referee 1: comments to the manuscript Novel photocatalytic nanocomposite made of

polymeric carbon nitride and metal oxide nanoparticles (443503)

The paper reports on the novel photocatalytic nanocomposite of polymeric carbon nitride and metal oxide nanoparticles. Authors have focused their attention on catalyst, which does not contain precious metals and scarce compounds and represents thus advantage for commercial application. I have following comments/amendments:

R1-1: The adsorption and catalytic degradation experiments require more detailed and corrected, description:

The sentence ...In a 100 cm3 glass beaker 0.02 g/dm3 of the appropriate photocatalyst was

placed and then 100 cm3 of the dye solution or phenol at an initial concentration of 20 mg/dm3

was added.. .(row 274) is confusing, unclear and should be clarified what was the final

concentration of catalyst in a mixture? ...placing 0.02g/dm3 catalyst into 100 cm3 glass be says nothing about its amount and final concentration. What was the state of catalyst suspension in water? What was the final volume of suspension?

Answer to R1-1: We have put more details to our experimental procedure as recommended. The sentence in lines 277-279 was replaced to: In a 150 cm3 glass beaker 20 mg of the appropriate photocatalyst was suspended and then 100 cm3 of the dye or phenol solution with an initial concentration of 20 mg/dm3 was added. The final concentration of the catalyst in a mixture was 0,02 g/dm3. Lines: 420-443.

R1-2:Was the suspension stirred during adsorption and photodegradation steps? How?

Answer to R1-2: The following sentence was added to the experimental section: The suspension was stirred continuously using a magnetic stirrer (500 rpm). Lines: 420-443.

R1-3: How was the mixture irradiated? Note, that the solution column absorbs UV light as well, thus

its intensity at catalyst particles in the bulk is lower than at the slurry (suspension) surface. What was the intensity of the light at the suspension surface and how decays in the depth? (depth penetration of light affects the efficiency of photoreaction, which thus prefers thin layer arrangement).

Answer to R1-3: The dye or phenol solution with added photocatalyst was transparent for light. It was not typical suspension with homogeneously spread nanoparticles. However, the intensity of UV light was not checked in the bottom of the reactor. We will take this into consideration in our further studies. The radiation intensity on the solution surface was 30 W/m2 UV and 89 W/m2 Vis.

R1-4: The sentence “high intensity of UV light” (row 272) says nothing about its power and thus the

process efficiency cannot be evaluated. What was the wavelength of the UV light? All above mentioned details may significantly affect photodegradation efficiency.

Answer to R1-4: We added the following sentence to the manuscript as requested: The suspension was continuously radiated with the UV light source (6 lamps, 20 W each, 40 cm long, Koninklijke Philips N.V., Amsterdam, Netherlands) for 5 hours. Lines: 442-443.

R1-5: Did authors evaluate products of photodegradation of their model contaminants (phenol and

Orange II)? Photodegradation may result in toxic products as well, depending on the time of exposition and composition of solution.

Answer to R1-5: Degradation of the main pollutants (Orange II dye and phenol) was only studied.

R1-6: Did authors examine the wettability of the catalyst? The existence of ambient gaseous (micro)pockets in the heterogeneous mixture limits the area of the reaction interface and thus affects significantly the extent of both adsorption and degradation step.

Answer to R1-6: The wettability of tested photocatalysts was not examined. However, it can be clearly stated from Fig. 1a (IR measurement) the disappearance of hydroxyl groups. This means that the changes in surface wettability are possible to occur.

We also performed additional tests for pores evaluation. Additional explanation is provided on page no 4. We concluded that there is no influence of porosity on photocatalytic properties of nanocomposites.

R1-7: At least elemental statistics of photodegradation data (repeatability) is missing, which may reveal possible fluctuations caused by experimental arrangement.

Answer to R1-7: It has to be taken into consideration that the amount of the photocatalyst used was 20 mg (0,02 g/dm3), while the solution volume was 100 cm3. In order to check whether the obtained results are characterized by high repeatability, each test has to be conducted at least twice. Unfortunately, due to the complexity of the whole research and limited amount of obtained nanomaterials, all tests were carried out only once.

R1-8: The English requires moderate correction. Conclusion: The manuscript requires major revision

prior to final publication decision.

Answer to R1-8: The English was corrected as recommended.

Reviewer 2 Report

In this manuscript  a series of PCN/AlOOH/ZrO2-nanocomposites were synthesised, characterised and tested towards the photocomposition of phenol and Orange II. Synthesis and characterization were described in detail and correctly executed. The work is interesting, but the following remarks should be taking into account before the publication on Molecules:  

- Lines 121-122: Authors wrote “The change of the density values is related to applied annealing temperature…”, but the density values are the same in three samples and the only different i the sample annealed at 450°C. Please reformulate the sentence and give a hypothesis for this trend.

- Figure 4: The use of different time scales for the data in graphs 4a, 4b and 4c makes the data not easy comparable. Please standardize the x-scale in all figures.

- Figure 5: The same as in figure 4.

- Figure 4 and 5: What is the utility to show the absolute change of phenol and Orange II concentration? Are the differences with respect to C/C0 significant?

- Lines 195-196: The authors  attribute the formation of PCN at 400°C, instead of 600°C, to “the presence of nano-sized metal oxide in the nanocomposite and fast heating rate (50 °C/min)”. What evidence supports this hypothesis? Have the authors performed tests at different heating rates to demonstrate their theory?

- Lines 223-225: The authors should better explain the properties of their material with respect to the cited examples. 

- Line 279: Please report the power of the UV-lamp, and eventually the wavelength emission range. 

Author Response

Answers to the Referee 2:

In this manuscript a series of PCN/AlOOH/ZrO2-nanocomposites were synthesised, characterised and tested towards the photocomposition of phenol and Orange II. Synthesis and characterization were described in detail and correctly executed. The work is interesting, but the following remarks should be taking into account before the publication on Molecules: 

R2-1: - Lines 121-122: Authors wrote “The change of the density values is related to applied annealing temperature…”, but the density values are the same in three samples and the only different in the sample annealed at 450°C. Please reformulate the sentence and give a hypothesis for this trend.

Answer R2-1: We decided to run some additional tests and modify paragraph related to BET and density. These are lines 118-172.

The relationship between SSA BET and density values for homogeneus nanomaterials was observed in our lab before. Please check the articles cited below for ZnO, ZrO2 and hydroxyapatite (1-3). It was observed that the highest BET the smallest density. Our PCN nanocomposites do not show mentioned trend (Table 1). In this case SSABET differ while density remains approximately constant. We attribute that lack of significant change of the density to applied annealing temperature, melamine condensation and its interconversion processes. It means for example, that for nanocomposite prepared at 450 oC nanoparticles and pores of Al2O3-ZrO2 are covered by PCN layer (closed pores) which causes lower density (helium pycnometry). At higher temperatures the PCN undergoes further interconversion and decomposition, which increases SSABET while density stays at the same value. Interestingly, the density values of nanocomposites annealed at 400, 500 and 600 oC are nearly the same. Only one sample (nanocomposite annealed at 450 oC) has slightly lower density (Table 1.). In addition, all prepared nanocomposites show smaller density than reference sample with density 3.75 g/cm3 (Al2O3-ZrO2 annealed at 400 oC). This fact could be explain by reduction of Zr4+ and Al3+ ions in a presence of melamine decomposition products and obtaining metallic precipitations not visible during XRD examination. Thus, these ions could lead to increased density of PCN-nanocomposites in comparison to the reference sample.

1. Opalinska, A.; Malka, I.; Dzwolak, W.; Chudoba, T.; Presz, A.; Lojkowski, W. Size-dependent density of zirconia nanoparticles. Beilstein J. Nanotechnol. 2015, 6, 27-35. doi: 10.3762/bjnano.6.4

2. Wojnarowicz, J.; Chudoba, T.; Koltsov, I.; Gierlotka, S.; Dworakowska, S.; Lojkowski,W. Size control mechanism of ZnO nanoparticles obtained inmicrowave solvothermal synthesis. Nanotechnology 2018, 29, 065601. doi: 10.1088/1361-6528/aaa0ef

3. Kusnieruk, S.; Wojnarowicz, S.; Chodara, A.; Chudoba, T.; Gierlotka, S.; Lojkowski, W. Influence of hydrothermal synthesis parameters on the properties of hydroxyapatite nanoparticles. Beilstein J. Nanotechnol. 2016, 7, 1586–1601. doi:10.3762/bjnano.7.153

R2-2: Figure 4: The use of different time scales for the data in graphs 4a, 4b and 4c makes the data not easy comparable. Please standardize the x-scale in all figures. Figure 5: The same as in figure 4.

Answer to R2-2:  If you compare for example figure 4a, with its equivalent figure 5a the x-scale is the same. Similar in case of figures 4b,c and 5b,c. Authors decided not to change x-scales because either we will lose some important results (if the scale will be up to 3h) or there will be gaps on the figures which looks  unprofessionally (when the scales will be up to 6h).

R2-3: - Figure 4 and 5: What is the utility to show the absolute change of phenol and Orange II concentration? Are the differences with respect to C/C0 significant?

Answer to R2-3: Authors would prefer to keep all figures 4 and 5 without changes. There are visible differences between absolute changes of Phenol/Orange concentrations and relative change of pollutant concentration. This is apparent especially once you compare Fig. 5b and Fig. 5c.

R2-4: - Lines 195-196: The authors  attribute the formation of PCN at 400°C, instead of 600°C, to “the presence of nano-sized metal oxide in the nanocomposite and fast heating rate (50 °C/min)”. What evidence supports this hypothesis? Have the authors performed tests at different heating rates to demonstrate their theory?

Answer to R2-4: We did perform heating of pure melamine in various furnaces, heating rate and atmospheres (helium or air). In literature we can find information that PCN is formed 550-600oC temperature range when the sample is heated up with 5-10K/min heating rate. We obtained PCN in conditions from literature, but PCN-Al2O3-ZrO2 nanocomposites were not formed in those conditions. Also it should be mentioned that pure melamine heated up in tube furnace with heating speed 50K/min up to 400oC did not result in pure PCN. That is why we think that there are two main factors influencing PCN-nanocomposite formation: heating rate 50K/min and suitable nanometric surface of AlOOH-ZrO2 nanopowders.

R2-5: - Lines 223-225: The authors should better explain the properties of their material with respect to the cited examples.

Answer to R2-5: the changes were made as requested. Lines 320-347.

R2-6: - Line 279: Please report the power of the UV-lamp, and eventually the wavelength emission range.

Answer to R2-6: The changes were made as requested: lines 422-443:

The suspensions were then irradiated with ultraviolet light (6 lamps, 20 W each, 40 cm long, Koninklijke Philips N.V., Amsterdam, Netherlands) for 5 h.

Round  2

Reviewer 1 Report

Just adding wavelength of UV light used in experiment is still necessary for completion.

Then paper is ready for publication.

Author Response

The information was added to the manuscript (lines 396-399):

Photocatalytic activity of the obtained nanomaterials was determined on the basis of water pollutants degradation tests under the influence of UV-Vis radiation with high UV intensity (6 lamps, 20 W each, 40 cm long, type: ISOLDE, Koninklijke Philips N.V., Amsterdam, Netherlands). The lamp used in experiments is not monochromatic. The wavelength range is from 200-800 nm.